# The Haystack Telescope as an Astronomical Instrument

**Jens Kauffmann** *⁕ , **Ganesh Rajagopalan** , **Kazunori Akiyama** , **Vincent Fish** , **Colin Lonsdale** , **Lynn D. Matthews** and **Thushara Pillai**

Massachusetts Institute of Technology Haystack Observatory, 99 Millstone Rd., Westford, MA 01886, USA
⁕ Correspondence: jens.kauffmann@mit.edu

**Abstract:** The Haystack Telescope is an antenna with a diameter of 37 m and an elevation-dependent surface accuracy of ≤100 μm that is capable of millimeter-wave observations. The radome-enclosed instrument serves as a radar sensor for space situational awareness, with about one-third of the time available for research by MIT Haystack Observatory. Ongoing testing with the K-band (18–26 GHz) and W-band receivers (currently 85–93 GHz) is preparing the inclusion of the telescope into the Event Horizon Telescope (EHT) array and the use as a single-dish research telescope. Given its geographic location, the addition of the Haystack Telescope to current and future versions of the EHT array would substantially improve the image quality.

**Keywords:** Very Long Baseline Interferometry; radio astronomy; millimeter astronomy; radio telescopes; high angular resolution; astronomical instrumentation

## 1. Introduction: Astronomy Observations with the Haystack Telescope

MIT Haystack Observatory has been a home to a radome-enclosed telescope of 37 m diameter since 1964 [1][1]. Figure 1 illustrates the siting of the instrument, while Figure 2 presents an overview of the dish. The original system was primarily conceived as a space radar and as a platform for telecommunications experiments to support work by MIT Lincoln Laboratory. Ownership was transferred to the Northeast Radio Observatory Corporation[2] (NEROC) in 1970, with the goal to enable millimeter-wave observations for the astronomy community in the Northeast US, while still being available as a radar sensor to MIT Lincoln Laboratory. The site is known as MIT Haystack Observatory since that time. The telescope has undergone several upgrades since its original dedication. Some of this work focused on improving the surface accuracy of the dish, which was improved from an initial root-mean-square (RMS) error of ∼900 μm to ∼200 μm after 1992 [2].

The Haystack Telescope has enabled key scientific discoveries, as summarized by Whitney et al. [3]. Radar observations delivered key intelligence on the Apollo landing sites, and joint observations with the Westford Telescope—a dish of 18 m diameter located about a mile from the Haystack Telescope—produced the first radar maps of Venus' surface that cleanly separated radar echoes from the planet's northern and southern hemispheres. Radar observations of Venus and Mercury also delivered stringent tests of General Relativity by constraining the gravitational time delay caused by the presence of the Sun (i.e., Shapiro delay, [4]). Single-dish spectroscopy observations with the Haystack Telescope were essential in establishing "dense molecular cores" as the key star-forming sites in molecular clouds [5], and they showed how dense cores build up density as they contract out of more diffuse cloud material [6]. MIT Haystack Observatory led the way during the inception of Very Long Baseline Interferometry (VLBI), and 8 of the 22 awardees of the 1971 Rumford Prize of the American Academy of Arts and Sciences for the inception of VLBI were working at the observatory. The Haystack Telescope critically supported this work. VLBI observations with the instrument, such as the discovery of apparent superluminal motion in quasars [7], shaped our understanding of the universe at high angular resolution.

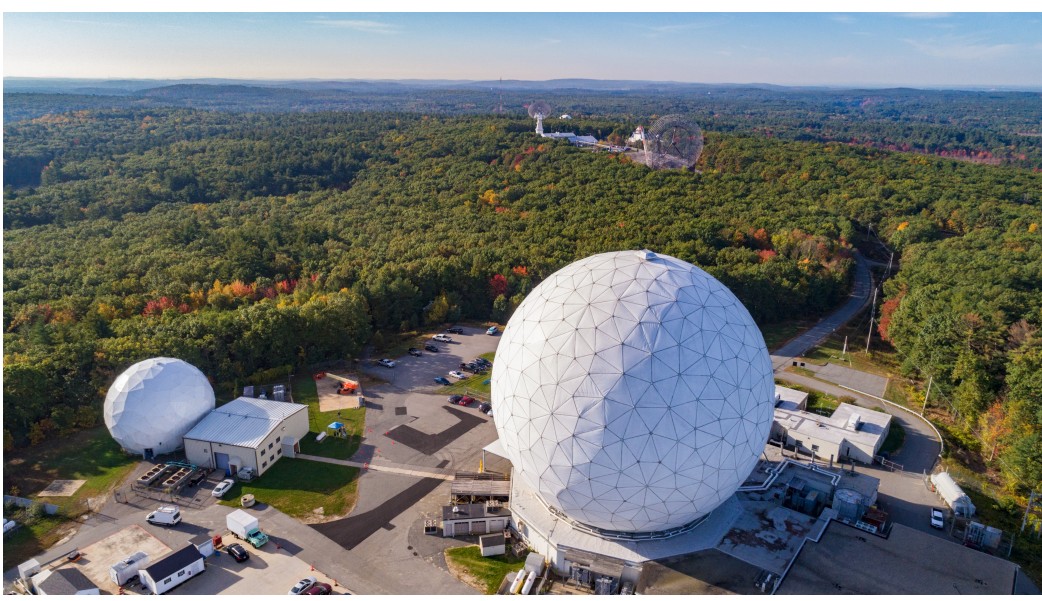

**Figure 1.** Aerial view of MIT Haystack Observatory. The Haystack Telescope, a dish of 37 m diameter, is located in the large radome dominating the foreground. (Used with permission, courtesy of Mark Derome).

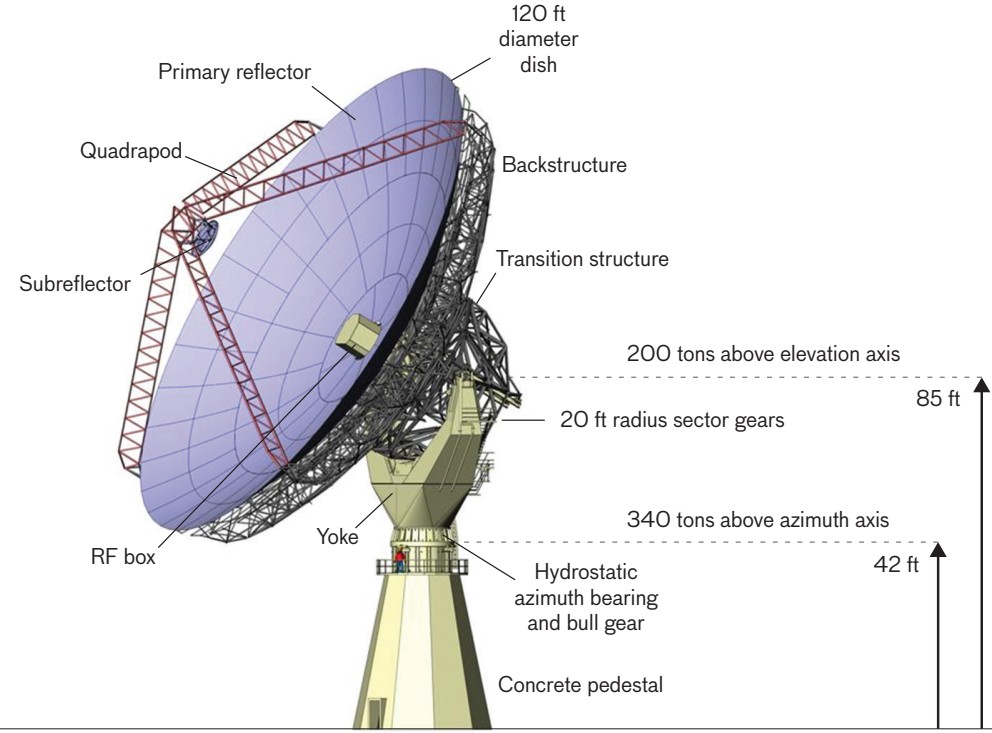

**Figure 2.** Overview of the Haystack Telescope, with the radome removed [8]. The receiver equipment is installed in the "RF box", a container that is brought down to ground level during "box-down" periods to enable major engineering activities. (Reprinted with permission, courtesy of MIT Lincoln Laboratory, Lexington, MA, USA).

A major upgrade, completed in 2014, improved the surface accuracy to $\leq$100 μm, depending on elevation. This work, executed by MIT Lincoln Laboratory under sponsorship by the Defense Advanced Research Projects Agency (DARPA) and the U.S. Air Force, was part of the upgrade delivering the Haystack Ultrawideband Satellite Imaging Radar (HUSIR). The HUSIR system is designed around a W-band radar covering a substantial

bandwidth of 92–100 GHz, and it also includes an X-band radar operating at 9.5–10.5 GHz. The outstanding bandwidth of $\Delta \nu = 8$ GHz enables the W-band radar to directly resolve structures of $c/(2 \cdot \Delta \nu) = 1.9$ cm size in range [9], with advanced image processing techniques delivering an effective resolution well below this scale. In 2014, HUSIR's W-band radar delivered the finest spatial resolution of any imaging radar, while the X-band radar constituted the only system for imaging out to geosynchronous orbits [9]. The systems have been upgraded since, and HUSIR continues to be an essential contributing sensor for space situational awareness.

Today, NEROC has access to about one-third of the time available on the Haystack Telescope. This time can be used to conduct experiments in astronomy and other fields of fundamental research. The primary access windows are weekends, and night hours at 23:00–07:00 local time on Mon.–Fri. Access to other periods, as for example needed for time-critical experiments in astronomy, is coordinated with MIT Lincoln Laboratory. Such work can currently use K-band (18–26 GHz) and W-band receivers (85–93 GHz) dedicated to astronomical observations that are separate from the HUSIR systems. An existing Q-band system covering 36–50 GHz will be brought online in the future.

MIT Haystack Observatory currently studies the expected performance of the telescope at ∼230 GHz. This is done as part of the ngEHT project (as described elsewhere in this special issue; also see https://www.ngeht.org, accessed on 2022 Dec. 15), which seeks to deliver a "next generation EHT" by adding new stations and other capabilities to the Event Horizon Telescope (EHT; see [10] for a recent description of the system). Inclusion of the Haystack Telescope into the EHT would enhance the imaging capabilities of the array, as described below.

The upgraded dish provides exciting opportunities for astronomy. Unfortunately, until recently it was not possible to make use of the telescope's capabilities, given the lack of substantial and systematic funding for astronomical experiments. This has changed in the past few years, thanks to a private donation and a grant from the National Science Foundation supporting the ngEHT project (AST-1935980). The telescope is currently regularly used to conduct experiments in support of system commissioning and initial experiments into astrophysical research and education. This includes three VLBI runs at 86 GHz that have delivered fringe detections on intercontinental baselines.

This paper is organized as follows. The telescope, its current and future instrumentation, and the characteristics of the site are described in Section 2. The discussion in Section 3 outlines the case for research, education, and technology development on the Haystack Telescope. The connection of the telescope to the EHT and ngEHT projects is described in Section 4. The material is summarized in Section 5.

## 2. Telescope, Instrumentation, and Site
### 2.1. Telescope and Site

Figure 2 summarizes the characteristics of the dish. The reflector of the Haystack Telescope has a diameter of 120 ft, equivalent to 36.57 m. It is formed by 432 panels that each have an RMS surface accuracy of about 28 μm [8]. The main reflector itself is rigged to achieve am RMS surface accuracy of 75 μm at an elevation of 25°, with larger deformations occurring at higher or lower elevations [11]. The moving sections have a mass of 340 t, with 200 t of mass moving in elevation. The dish is capable of slewing at speeds of $5° \, \text{s}^{-1}$ in azimuth and $2° \, \text{s}^{-1}$ in elevation, and it achieves accelerations of $1\overset{\circ}{.}5 \, \text{s}^{-2}$ and $2° \, \text{s}^{-2}$, respectively. By requirement, the pointing accuracy is $< 3\overset{''}{.}6$, with a tracking accuracy $< 1\overset{''}{.}8$ [11].

The telescope is housed in a radome of 150 ft diameter that was originally designed for use in extreme arctic environments and is capable of withstanding 130 mph winds (i.e., $210 \, \text{km} \, \text{h}^{-1}$ or $60 \, \text{m} \, \text{s}^{-1}$) [8]. The radome is skinned with three-ply ESSCOLAM 10 membranes with a hydrophobic coating, which are characterized by a transmission of about 95% at 90 GHz [8].

The receivers are housed in a "box" that is installed about 85 ft above ground. It can be brought down to the floor of the telescope building during dedicated "box-down" periods. The box houses radar equipment as well as the astronomy receivers, and it is very tightly packed with systems. As a consequence, major engineering actives can only be performed during a box-down window, during which the interior of the box can be accessed easily from all sides. The number and duration of box-down periods is minimized in support of high-priority radar observations.

The observatory's land is distributed over the Massachusetts towns of Groton, Tyngsborough, and Westford, with Westford being the administrative home of MIT Haystack Observatory. This thickly forested community is about an hour's drive away from downtown Boston (MA). The telescope itself is located at $42°.62$ N vs. $71°.49$ W, at an altitude of 130 m.

Haystack Observatory experiences extended periods of cold and dry weather during the winter, thus providing the weather conditions needed for observations at millimeter wavelengths. Historical measurements of the precipitable water vapor (PWV) column are available from the Suominet[3] network for atmospheric research. Archived data give a median PWV column of 8.3 mm for the period November 1 to April 30. Assuming an outside temperature of 0 °C, modeling of the atmosphere with the AM[4] radiative transfer code gives a corresponding optical depth of 0.12 at ∼86 GHz under such conditions, equivalent to an atmospheric transmission of 84% at 45° elevation. More realistically, observations by systems like the EHT are triggered in better-than-median atmospheric conditions. To give an example, the PWV column is below 5.3 mm for 25% of the winter period. Rich additional documentation about the telescope and the site can be found in Brown and Pensa [1], Whitney et al. [3], Waggener [8], Czerwinski and Usoff [9], Usoff et al. [11], MacDonald et al. [12], and Eshbaugh et al. [13].

### 2.2. Current Instrumentation

The telescope is equipped with receivers operating in the K (18–26 GHz), Q (36–50 GHz), and W bands (70–115 GHz). The cryogenic frontends operate at around 20 K in independent dewars. These are arranged roughly on a vertical line that is offset from the central focal point. MIT Lincoln Laboratory operates on-axis X-band and W-band radars, so that the three astronomy receivers are offset from boresight. The sub-reflector on a hexapod is remotely controlled to choose between the three K, Q and W-band receivers. Current observing projects make use of the K and W bands, and these receivers are therefore currently kept operational by engineering activities.

The K-band frontend is shared between MIT Haystack Observatory and MIT Lincoln Laboratory. One polarization is available for astronomical observations, while the other polarization is used for holography observations. Astronomical observations can be conducted anywhere in the frequency range of 18–26 GHz.

The W-band frontend is currently configured as a single-sideband receiver that senses horizontal and vertical polarization. Data are taken in a sideband of 8 GHz width that is set by an analog bandpass filter. The system is currently set up to observe at frequencies of 85–93 GHz. Modest upgrades to the hardware would allow to access the full frequency range of 70–115 GHz. The receiver was recently improved via the installation of a new wideband low-noise amplifier (LNA) and components for the sideband rejection scheme. These investments were made possible by an NSF MSRI-1 grant to the ngEHT project (AST-1935980).

The backends are located at the ground level of the telescope building. A radiofrequency-over-fiber (RFoF) system is used to transport the signals into this room. The RFoF infrastructure is currently being upgraded for transport bandwidths of up to 20 GHz for two polarizations. An up-down converter (UDC) is used to condition the signals for the backends. The single-dish backend currently processes up to 500 MHz in one polarization. Further investments in hardware and software would enable processing of larger bandwidths and of a second polarization. The backend measures continuum signals, and it currently also

produces spectra of up to 500 Hz resolution. VLBI data are acquired using a ROACH2 digital backend (R2DBE) connected to a Mark 6 VLBI recorder. A Rakon Oven Controlled Crystal Oscillator (OCXO) is used as a frequency standard for ongoing engineering experiments in VLBI. The acquisition of the RFoF infrastructure, and the ongoing acquisition of a new OCXO, are supported by an NSF MSRI-1 grant to the ngEHT project (AST-1935980).

### 2.3. Current and Future Instrument Development

Current work on the Haystack Telescope focuses on evaluation of the newly upgraded system (i.e., after installation of the W-band LNA, RFoF system, and VLBI equipment). While characterization of the W-band system is the main activity, the K-band receiver is occasionally used to deliver complementary data on telescope performance under less ideal weather conditions. This program consists of single-dish observations of calibrators like planets, as well as participation in observations by VLBI networks. In the area of interferometry the goal is to enable future VLBI observations at $\lesssim$90 GHz, and to assess the feasibility of VLBI observations at $\sim$230 GHz in support of the EHT. More generally, the observations seek to demonstrate the capability of the Haystack Telescope to deliver exciting astrophysical research as a single-dish telescope and VLBI station.

Current funding from an NSF MSRI-1 grant (AST-1935980) supports the design of a receiver for VLBI observations with the Haystack Telescope at $\sim$230 GHz in the context of the ngEHT project. This undertaking might evolve into the design for a multi-band receiver enabling parallel observations at $\sim$86 GHz and $\sim$230 GHz. This depends on future decisions by the EHT and ngEHT projects concerning the need for multi-band observations in support of "frequency phase transfer" (FPT, [14]), i.e., the transfer of VLBI calibration information obtained at one frequency to other bands.

The long-term plan for the telescope foresees to support single-dish and VLBI observations at K, Q, and W band, as well as at $\sim$230 GHz. Ongoing investigations will clarify whether some or all of these receivers need to be able to observe in parallel (e.g., to support FPT). The installation of wideband (i.e., $\geq$8 GHz) receivers and backends for single-dish and VLBI observations, as well as of a maser clock as a frequency standard for VLBI experiments, form part of this long-term plan. The development of the telescope must be supported via dedicated grants from funding agencies, as the Haystack Telescope receives no general-purpose funding to advance the capabilities of the facility.

## 3. Astrophysical Research, Education, and Technology Development

Section 4 explains how the Haystack Telescope can add new, sensitive, and important baselines to the EHT. In the same way, the Haystack Telescope can complement the Global Millimeterwave VLBI Array (GMVA). The increased availability of multi-band receivers on radio observatories, as pioneered by the Korean VLBI Network (KVN) [15], raises the exciting prospect for the Haystack Telescope to join an intercontinental network building on FPT.

The outstanding scientific capabilities of large single-dish instruments at millimeter wavelengths are demonstrated by the high impact of current research on the IRAM 30m-telescope. Recent work with that telescope includes the study of star formation physics in nearby galaxies [16], and investigations of molecular cloud structure and evolution in the Milky Way that support the aforementioned extragalactic work [17,18]. Many of these studies employ the EMIR receiver system that samples 16 GHz of bandwidth per polarization in the 70–115 GHz range [19]. Installation of receiver and backend systems matching or exceeding this capability would open up new and exciting capabilities for the US-based community.

The Haystack Telescope offers unique, important, and currently missing educational opportunities in the US. The instrument is associated with the NEROC community of 13 research-intensive educational institutions in the Northeast US (see footnote 5 on page 8). Several of these are closely involved in the EHT, the Large Millimeter Telescope (LMT), and the Submillimeter Array (SMA). The Haystack Telescope is within easy driving dis-

tance from all these institutions, providing outstanding hands-on experiences for junior researchers within NEROC. More generally, the telescope can serve as a destination for educational astronomical daytrips that are independent from daytime, and only modestly dependent on the weather, by schools, colleges, and universities in six states of the US (all of MA, RI, CT; most of NH; parts of ME and NY)[5].

Operations of the Haystack Telescope are exclusively funded by grants with specific objectives and deliverables. MIT Haystack Observatory receives no general long-term funding that could make the instrument broadly accessible by the community. Future operations are expected to be supported via a mix of funding streams. This will include observations for specific user groups that will pay for having their data taken on the Haystack Telescope. The current engineering work undertaken on an NSF MSRI-1 grant supporting the ngEHT project constitutes one example for such observations. Similarly, NSF AAG grants could fund data collection for specific astrophysical research projects. It is the ambition of MIT Haystack Observatory to also make the telescope broadly available to the entire US community. The feasibility of such a program depends on the grants acquired by the observatory.

### 4. Connection to the Event Horizon Telescope: Current Work and Future Roles

Ongoing work on the Haystack Telescope is in part funded by an NSF MSRI-1 grant (AST-1935980). This award forms part part of the ngEHT project, and its goal is to evaluate the telescope for inclusion into the EHT at $\sim$230 GHz via quantitative modeling and VLBI test observations at $\sim$86 GHz. A private donation to MIT Haystack Observatory enables parallel activities that enhance the overall capabilities of the instrument.

Figure 3 shows that addition of the Haystack Telescope to future versions of the EHT network would produce new and critical baselines. This is specifically demonstrated here for a network that includes the telescopes that are now available for future EHT observations, but that also includes a set of additional antennas enhancing the EHT. This can be seen in Figure 3 (top right), where dishes enhancing the current EHT constellation are marked by green stars. The resulting significant improvements to the *uv*-coverage of the array can result in reductions of the inner sidelobes of the synthesized beam by a factor 1.4 (K. Akiyama, priv. comm.). This is indicated by imaging simulations assuming the reference array summarised in Figure 3 (top left). The simulation in particular demonstrates that the addition of the Haystack Telescope to the EHT array would substantially improve the sampling of the *uv*-domain at baselines of $\lesssim 4 \times 10^9 \, \lambda$ (Figure 3 bottom), resulting in the aforementioned reduction of sidelobes. Data from the Haystack Telescope can also improve other practical aspects of VLBI observations. For example, the dish can deliver an important connection between dishes in Europe and the Americas. The telescope can also add substantial sensitivity to VLBI networks, in particular at low frequencies where the telescope efficiencies are higher. All these factors aid in the overall calibration of VLBI networks, delivering advantages beyond the fundamental improvement in *uv*-coverage. Importantly, the specific model shown here demonstates that that the Haystack Telescope would still add substantial value to the EHT array when considering network configurations that include more telescopes than those used today. The ngEHT project is developing reference arrays for future quantitative array assessments by the collaboration. Such evaluations will in future also include the Haystack Telescope.

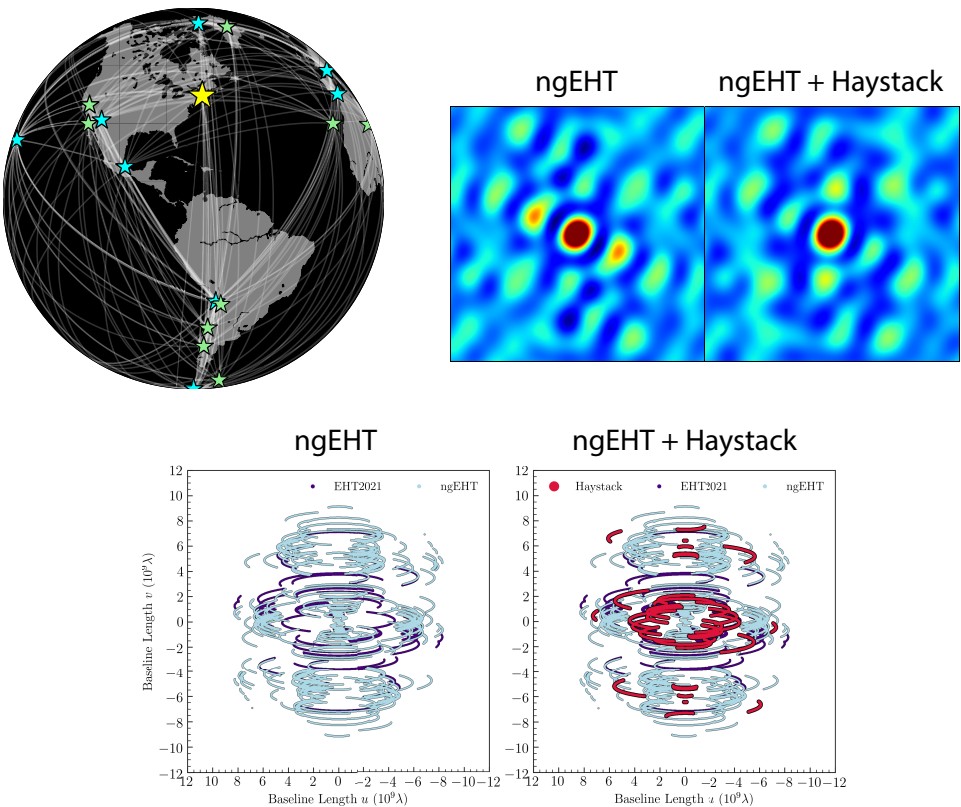

**Figure 3.** Outline of the impact of observations with the Haystack Telescope on VLBI arrays (K. Akiyama, priv. comm.). The **upper left panel** illustrates an ngEHT reference array used for imaging simulations, as appropriate for a target at a declination of $+10°$. Specifically, current EHT stations are marked by blue stars, while green stars indicate potential new sites. The Haystack Telescope is marked by a yellow star. This highlights the important role the Haystack Telescope can play in linking VLBI stations across the Atlantic Ocean. The **upper right panel** characterizes how inclusion of the Haystack Telescope into the adopted ngEHT array improves the synthesized beam. Inner sidelobes are reduced by a factor 1.4. The **bottom panel** illustrates that adding the Haystack Telescope would in particular help to populate the inner area of the *uv*-plane.

The Haystack Telescope would add substantial sensitivity to the EHT array. Consider observations at 0 °C outside temperature and a better-than-median winter PWV column of 5.3 mm (Section 2.1). In that case the atmospheric transmission is 64% at 230 GHz and $45°$ elevation. Preliminary performance modeling (J. Kauffmann, priv. comm.) further indicates an aperture efficiency of 35% and a radome transmission of 73%. Multiplication of all these factors shows that the telescope's effective combined efficiency is 16%. This number is small—but the effective aperture of this telescope is still equivalent to an *ideal* dish of $0.16^{1/2} \cdot 37$ m = 15 m diameter above Earth's atmosphere. This equal to the median dish size of current EHT stations[6], and smaller telescope diameters are considered for some future EHT stations. This underlines the role which the Haystack Telescope can play within the future EHT network. That said, this calculation *purely* considers the transmission losses of the system: the impact of ground-pickup and sky brightness on the system temperature are not included, given insufficient modeling at this time, while impact of the atmospheric transmission in the calibration to the $T_A^*$-scale *is* taken into account. For reference, repetition of the analysis for 86 GHz frequency and $45°$ elevation yields an equivalent diameter of 31 m for an ideal telescope above Earth's atmosphere. At this frequency the Haystack Telescope could serve as an "anchor station" that can be used to improve the overall calibration of the network. In particular, the instrument could serve this purpose at ∼90 GHz in support of FPT to smaller dishes (see Issaoun et al., this volume).

A key component of ongoing work is to validate the telescope's abilities via participation in VLBI runs conducted at ∼86 GHz frequency. The Haystack Telescope has joined three such experiments since April 2022. This has already resulted in the detection of fringes on intercontinental baselines. Ongoing analysis will quantitatively characterize the value of the Haystack Telescope in VLBI arrays.

## 5. Summary

The reflector of the Haystack Telescope has been upgraded to a dish of 37 m diameter that has a surface accuracy of ≤100 μm, depending on elevation (Section 1). The instrument serves as a radar sensor for space situational awareness, with about one-third of the time available for research by MIT Haystack Observatory. Current work funded by an NSF MSRI-1 grant conducts astronomical single-dish and VLBI observations at frequencies of ∼20 GHz and ∼90 GHz to study the inclusion of the telescope into the EHT array. Parallel work enabled via a private donation generally enhances the capabilities of the instrument for research and education. The telescope is housed in a radome of 150 ft diameter that is designed to support radar observations at high frequency (Section 2). Current data indicate a median precipitable water vapor (PWV) column of about 8 mm during winter months (i.e., November 1 to April 30). These characteristics enable the Haystack Telescope to provide the US-based community with new and important capabilities for research, education, and technology development in radio astronomy (Section 3). In particular, the instrument can add new transatlantic baselines to the EHT network that would drastically improve the image quality with a frequency-dependent dish sensitivity equivalent to an ideal telescope of 15 m to 31 m above Earth's atmosphere (Section 4). Initial VLBI experiments conducted in April 2022 have resulted in fringe detections on intercontinental baselines.

**Author Contributions:** Conceptualization, J.K. and G.R.; writing—original draft preparation, J.K. and G.R.; writing—review and editing, K.A., V.F., C.L., L.D.M. and T.P.; visualization, K.A.; funding acquisition, L.D.M. and V.F. All authors have read and agreed to the published version of the manuscript.

**Funding:** This work was in part enabled by grants from the National Science Foundation, including DUE-1503793 and AST-1935980. The development of the Haystack Telescope is further supported by a private donation.

**Data Availability Statement:** Not applicable.

**Conflicts of Interest:** The authors declare no conflict of interest.

## Abbreviations

The following abbreviations are used in this manuscript:

| | |
|---|---|
| EHT | Event Horizon Telescope |
| LMT | Large Millimeter Telescope |
| ngEHT | next generation Event Horizon Telescope |
| SMA | Submillimeter Array |
| VLBI | Very Long Baseline Interferometry |

## Notes

[1] This article includes numerous references to the "Celebrating 50 Years of Haystack" Special Issue of the Lincoln Laboratory Journal, which is available at https://www.ll.mit.edu/about/lincoln-laboratory-publications/lincoln-laboratory-journal/lincoln-laboratory-journal-0 (accessed on 15 December 2022).

[2] The current NEROC members are Boston College, Boston University, Brandeis University, Dartmouth College, Harvard University, Harvard-Smithsonian Center for Astrophysics, Massachusetts Institute of Technology, Merrimack College, University of Massachusetts at Amherst, University of Massachusetts at Lowell, University of New Hampshire, and Wellesley College. NEROC's mission is to further research, education, and scientific collaboration in the field of radio science. NEROC is headquartered at MIT Haystack Observatory. Also see https://www.haystack.mit.edu/about/northeast-radio-observatory-corporation-neroc/ (accessed on 15 December 2022).

3    https://www.cosmic.ucar.edu/what-we-do/suominet-weather-precipitation-data (accessed on 15 December 2022)

4    https://lweb.cfa.harvard.edu/~spaine/am/ (accessed on 15 December 2022)

5    Permitting a one-way drive time of $\leq 3$ h, following https://www.smappen.com/app/ (accessed on 15 December 2022).

6    The median dish diameter of the EHT array available for future observation cycles is 15 m. This characterizes an array formed from the phased ALMA dishes, with a collection area equivalent to an antenna of 91 m diameter, the phased NOEMA dishes, equivalent to an antenna of 52 m, and the phased dishes of the SMA, equivalent to an antenna of 17 m. The array also includes the LMT of 50 m diameter, the IRAM 30m-telescope, the JCMT of 15 m diameter, the APEX, Kitt Peak, and GLT dishes of 12 m diameter, and the SMT and SPT dishes of 10 m diameter.

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
