# Peer review of "The Haystack Telescope as an Astronomical Instrument"

_galaxies, doi:10.3390/galaxies11010009_

Round 1

Reviewer 1 Report

This paper is a concise and informative article describing the history, present capabilities and upgrade paths of the Haystack Telescope, with a focus on its possible role as part of the Event Horizon Telescope array. 

 I have a few small suggestions that may increase the strength of its argument (that Haystack should join the EHT). 

Figure 3 is key to the central theme and highlights the geographical position of Haystack with respect to other EHT antennas. However, the dual-panel comparison of the synthesised beam with and without Haystack requires close examination to spot the differences. The improved u,v coverage afforded by Haystack could be highlighted with an additional panel showing the u,v tracks over, say, 10 hours, or a histogram of baseline density vs length, with Haystack's contribution highlighted in a different colour or line style?

It may also be helpful to expand slightly on the practical benefits that result from improved u,v coverage. This is clear to an expert radio astronomer, but may be less obvious to others.

The potential sensitivity improvements of including Haystack in the EHT are discussed around line 232, where it may be helpful to expand on the number and typical size of existing EHT antennas and give Haystack's nominal collecting area as a fraction of the existing EHT total, to emphasise its relative contribution.

A few other very minor comments:

Line 44: HUISIR in brackets may be a typo, the acronym is HUSIR elsewhere

Line 193: "closely involved the" seems to be missing a word

Line 212: "my" should be "by"

Author Response

Thank you very much for your thoughtful comments. We believe that your input substantially improves the readability of the paper.   You suggested to improve the presentation in Fig. 3. We have now done so by including plots of uv-coverage with and without the Haystack Telescope. This illustrates more clearly which regions of the uv-plane will be populated better when including the Haystack Telescope into the EHT. We have also provided more detail about the nature of the reference array used here.   You requested to include a more comprehensive comparison of dish sizes within the EHT array. We are now doing this in footnote 6, where we derive a median dish size of 15m for EHT telescopes. We have also refined our discussion of the effective size of the Haystack Telescope (i.e., when considering efficiencies) under more realistic observing conditions.   We have also addressed the typos and grammatical mistakes identified by you.

Author Response

Thank you very much for your thoughtful comments. We believe that your input substantially improves the readability of the paper.     You voiced a concern that the EHT-specific parts of the paper are relatively brief, compared to the rest of the presentation. This results from a lack of recent documentation about the Haystack Telescope. We have made the experience that many of our colleagues want to know about the overall status of the facility, the overall (radar-driven) constraints under which the instrument can be operated as an astronomical research instrument, and our general future plans to use the telescope for research and education. The development of observational modes for the EHT is one component in all of this, but it should be presented as one component in a more comprehensive plan for the facility. We feel this is in particular warranted given that the telescope must be operated as a multi-purpose facility for astronomy, if we wish to raise the support needed for inclusion in the EHT.   The limited focus on the EHT is also a consequence of the limitations in quantitative studies characterizing the Haystack Telescope as a potential EHT station. Recent work on the telescope has focussed on delivering a well-characterized research telescope operating at about 90 GHz. We have made substantial progress with this work in 2022, and we are now starting to use this new information for more comprehensive assessments of what the telescope can deliver as a VLBI station at 230 GHz. In this paper we focus on the limited material that can be defended well on the basis of the current but limited experience. This body of work should improve substantially in 2023.     In the same context you voiced concern about the level of detail at which calculations in Sec. 4 and Fig. 3 are documented. One current limitation is the lack of well-defined reference configurations for the ngEHT array. Figure 3 represents a constellation that was discussed in the summer of 2022, when this publication was developed. The ngEHT project is now moving towards a more well-defined reference array, and we will use that array in future studies. We agree that the Blackburn et al. (2019) configuration would also have been a plausible foundation for our experiments. However, it would not have been a better one, given the range of constellations that have been discussed until very recently.   We highlight that the emerging ngEHT reference array likely includes far lesser telescopes than assumed in Fig. 3. This means that our calculations overestimate the abilities of an array that does not include the Haystack Telescope. The inclusion of this dish should therefore result in more substantial image improvements than outlined here.   We have tried to better explain in text and caption the nature of the reference array adopted here, and the preliminary nature of our calculations. We in particular stress that future calculations should be based on emerging well-defined reference arrays.   We have also included uv-plots in Fig. 3 to provide a different perspective on the improvements delivered by inclusion of the Haystack Telescope. This clarifies our claim about important new baselines being delivered by the Haystack Telescope.     Thank you for highlighting our lack of references to the ngEHT and EHT projects. Unfortunately, we are not aware of good overview publications for these efforts, as existing publications focus on specific scientific or technical aspects of the systems. We have included a reference to a publication documenting the 2017 EHT array. We are also including pointers to the project websites.   Most importantly, we are now including language referring to other articles in this special issue (which we have not seen at any level of detail). This should, e.g., readers finding this paper via ADS aware that there is a series of papers which they should read.     Concerning educational opportunities described in Sec. 3, you inquired whether we could provide additional quantitative information on the potential impact of the telescope. Thank you for pointing out that we should characterize this even further. We are at this moment not able to state anything beyond what we are providing now. It depends on where we wish to set priorities. We could, for example, host classes of ~20 students in several day-long events focused on hands-on experiences. We have also run open house activities where ~150 visitors get to see the observatory, and this could in future include additional activities involving the telescope.   Again, thanks for pointing out the need for quantitative information in this area. We have to develop this, fast.     Thank you very much for your detailed look on language and spelling. We have, to our best knowledge, addressed the issues you raised.     You inquired whether the 1.9 cm radar resolution is dependent on additional factors. In particular, given statements in the Czerwinski et al. paper from which we cite, you inquired whether there is a dependence on distance. This is actually not the case. Czerwinski et al. do make a statement about the radar resolution, but they actually make a statement about REQUIREMENTS which the radar has to fulfill in order to be useful.
